# How is depth of anaesthesia assessed in experimental pigs? A scoping review

**Alessandro Mirra**[1,2]*, **Ekaterina Gamez Maidanskaia**[1], **Luís Pedro Carmo**[3,4], **Olivier Levionnois**[1‡], **Claudia Spadavecchia**[1‡]

**1** Anaesthesiology and Pain Therapy Section, Department of Clinical Veterinary Medicine, Vetsuisse Faculty, University of Bern, Bern, Switzerland, **2** Graduate School for Cellular and Biomedical Sciences, University of Bern, Bern, Switzerland, **3** Department of Clinical Research und Public Health (DCR-VPH), Vetsuisse Faculty, Veterinary Public Health Institute, University of Bern, Bern, Switzerland, **4** Norwegian Veterinary Institute, Ås, Norway

‡ OL and CS are joint senior authors on this work.
* alessandro.mirra@vetsuisse.unibe.ch

**Data Availability Statement:** All relevant data are within the paper and its Supporting Information files.

**Funding:** The authors received no specific funding for this work.

## Abstract

### Background

Despite the large number of pigs involved in translational studies, no gold standard depth of anaesthesia indicators are available. We undertook a scoping review to investigate and summarize the evidence that sustains or contradicts the use of depth of anaesthesia indicators in this species.

### Methods

Medline, Embase and CAB abstract were searched up to September 22$^{nd}$ 2022. No limits were set for time, language and study type. Only original articles of in vivo studies using pigs or minipigs undergoing general anaesthesia were included. The depth of anaesthesia indicators reported in the selected papers were divided in two categories: A, indicators purposely investigated as method to assess depth of anaesthesia; B, indicators reported but not investigated as method to assess depth of anaesthesia.

### Results

Out of 13792 papers found, 105 were included after the screening process. Category A: 17 depth of anaesthesia indicators were found in 19 papers. Studies were conducted using inhalant anaesthetics as the main anaesthetic agent in the majority of the cases (13/19 = 68.4%), while 3/19 (15.8%) used propofol. The most investigated depth of anaesthesia indicators were bispectral index (8/19 = 42.1%) and spectral edge frequency 95% (5/19 = 26.3%). Contrasting results about the specific usefulness of each depth of anaesthesia indicators were reported. Category B: 23 depth of anaesthesia indicators were found in 92 papers. The most reported depth of anaesthesia indicators were: motor response following a stimulus (37/92 = 40.2%), depth of anaesthesia scores (21/92 = 23.3%), bispectral index (16/92 = 17.8%) and spectral edge frequency 95% (9/92 = 9.8%).

**Competing interests:** The authors have declared that no competing interests exist.

## Conclusion

Results highlight the lack of scientifically valid and reliable indicators to ensure adequate depth of anaesthesia in pigs.

## Introduction

The porcine model is extensively used in translational medicine due to the anatomical and physiological similarities between humans and pigs [1–4]. The last European Union's Report on animal research estimated that between 2015 and 2017 around 80000 pigs were used yearly in biomedical research [5]. Despite the large number of animals involved in translational studies, no "gold standard" methodology has been defined to assess anaesthetic depth. The impact of this problem is even more pronounced when neuromuscular blocking agents (NMBAs) are administered [6, 7]. This is not only a serious ethical concern, but it raises questions about the reliability of the results obtained from translational studies, especially those assessing perioperative cardiovascular, respiratory and hormonal parameters.

We undertook a scoping review to investigate and synthetize the evidence that sustains or contradicts the use of the depth of anaesthesia (DoA) indicators reported in the literature and adopted in pigs and minipigs undergoing general anaesthesia. In order to reach this goal, the following research question was defined: "Is there scientific evidence of the usefulness of the methodologies commonly employed to assess DoA in pigs?".

A scoping review was judged appropriate to provide a comprehensive and thorough overview of the available literature in the broad subject of interest.

The aims of the present work were:

- To identify the indicators that have been investigated to assess DoA in pigs;

- To investigate and summarize the evidence that sustains or contradicts the use of DoA indicators reported in the literature and adopted in pigs and minipigs undergoing general anaesthesia;

- To identify knowledge gaps.

## Materials and methods

The present scoping review was conducted following the PRISMA Extension for Scoping Reviews (PRISMA-ScR) guidelines [8]. A protocol, adapted from the PRISMA-P guidelines [9] was developed *a priori* (see supplementary data). The search strategies were developed in collaboration with the systematic review services coordinator of the University of Bern.

### Stage 1: Identification of DoA indicators

<u>Phase 1</u>: an electronic literature search was performed on January 20[th] 2020 by a single investigator (AM) using three databases (PubMed, Embase, CAB abstract) via Ovid, to develop a preliminary list of DoA indicators reported in pigs (material available from the author upon request). No time, language and study type limitations were set. The identified citations were imported into the Mendeley Reference Management Software and duplicates were removed. Title, abstract and keywords were screened to identify DoA indicators. In case it was not clear

in which species a certain DoA indicator was used, it was included in the list. Moreover, additional searches were performed via Google Scholar.

Phase 2: an initial list was developed including all the DoA indicators found during phase 1. This list was provided to anaesthetists of the Anaesthesiology and Pain Therapy Section, Department of Clinical Veterinary Medicine, Vetsuisse Faculty, University of Bern, who were asked to check it and add potential missing items.

Phase 3: the developed list was sent world-wide through the veterinary anaesthesia list server (ACVA-List) to veterinary anaesthesia specialists, with the same request. The final list including all DoA indicators collected from phases 1, 2 and 3 was established and used for the next step (Stage 2).

## Stage 2: Search strategy

Based on the list settled in Stage 1, a search strategy was developed (see supplementary data). It aimed at identifying papers including pigs undergoing general anaesthesia and where at least one of the previously listed DoA indicators was reported.

The electronic literature search was performed using the same aforementioned databases via Ovid: PubMed, Embase, CAB abstract. The first search was performed on March 19th 2020, and then updated on September 22nd 2022. No time, language and study type limitations were set. The identified citations were imported into Mendeley Reference Management Software and duplicates were removed.

## Stage 3: Paper selection

Two independent investigators (AM, EGM) screened title and abstract of papers identified through the first search (until March 19th 2020) and selected those that met all the following four inclusion criteria:

1. reported original research;

2. performed using pigs or minipigs;

3. performed in vivo;

4. involved animals undergoing general anaesthesia.

The investigators periodically met to compare screening results. Differences were discussed and resolved by consensus. In case one of the aforementioned points was not evident when reading title and abstract, the paper was included for further screening.

As the selection identified a large amount of papers, a further inclusion criterion (i.e. that an indicator of DoA must be mentioned in the title or abstract) was added and applied by a single investigator (AM).

For the search update (from March 19th 2020 to September 22nd 2022), papers were selected by a single investigator (AM) applying the five inclusion criteria.

## Stage 4: Full text screening

The full-text screening was performed by a single investigator (AM). In case of doubt, a second investigator (CS or OL) was asked to screen the text independently, and the final result was obtained upon consensus. During this phase, papers were excluded if:

- the authors did not report any DoA indicator;

- the DoA indicators were not reported for at least two well distinct time points and/or drug combinations and/or drug concentrations and/or physiological conditions and/or surgical interventions;

- unconsciousness was not pharmacologically induced (e.g., electricity, carbon dioxide);

- only a minimum alveolar concentration (MAC) extrapolated from other studies was used to guide anaesthetic administration;

- the paper was found to be a duplicate, or at least one of the five abstract inclusion criteria was not fulfilled;

- only the abstract was available.

  From the included papers, DoA indicators were identified and categorized as follows:

- category A: indicators purposely investigated as method to assess depth of anaesthesia;

- category B: indicators reported but not investigated as method to assess depth of anaesthesia

### Stage 5: Data extraction

Data were tabulated (Microsoft Excel) including the following information: first author's surname, year of publication, journal, DoA indicator investigated/used. Moreover, for papers reporting category A DoA indicators, further information was retrieved: animals' weight, age, breed and sex, number of animals included in the study, whether statistical analysis was performed, main anaesthetic drug used, summary of the main results.

### Stage 6: Data synthesis

Following categorization of indicators and data extraction, appropriate descriptive statistics were used to summarize the results obtained for each extracted variable.

## Results

### Stage 1: Identification of DoA indicators

The first literature search identified a total of 14682 papers, reduced to 12720 after duplicate removal. A total of 29 DoA indicators were identified and used to develop the final search strings (see supplementary data).

### Stage 2: Search strategy

The first search string identified 12304 papers, reduced to 10346 after duplicate removal.
    The update search identified 1488 papers, reduced to 1288 after duplicate removal.

### Stage 3: Study selection

Abstract selection of the first search results (until March 19[th] 2020) brought to the inclusion of 3600 papers, reduced to 302 once the fifth inclusion criterion was added; abstract selection from the updated search resulted in the inclusion of 32 additional papers. No duplicates were found.

## Stage 4: Full-text screening and classification

Of the 334 papers, 229 were excluded (68.6%). Among the 105 remaining papers, 19 included 17 DoA indicators of category A, while 92 papers included 23 DoA indicators of category B (Figs 1 and 2 and Tables 1 and 2).

## Stage 5 and 6: Data extraction and synthesis

**Category A.** All the studies focused on pigs and were published between 1972 and 2022. Animals' age was reported in 11/19 papers (57.9%); all pigs were younger than 6 months. In the majority of the cases (10/19 = 52.6%), animals of both sexes were included. Overall, a median [interquartile range 25%, interquartile range 75%] of 12 [6, 16] pigs were included in each study. The majority of the investigations were conducted using at least one inhalant anaesthetic as the main anaesthetic agent (13/19 = 68.4%, among which eight used isoflurane); four reported the use of propofol (3/19 = 15.8%) (Table 3).

The most commonly investigated DoA indicator was the bispectral index (BIS) (8/19 = 42.1%) [10–17]. In the majority of the studies reporting its use, inhalational anaesthetics were administered (7/8 = 87.5%). In two papers, no significant differences in BIS values were found among clinically relevant DoA levels [11, 12]. Using similar study designs, opposite results were found in two investigations in pigs undergoing isoflurane, sevoflurane or desflurane anaesthesia [14, 16], where BIS values significantly decreased at increasing anaesthetic MAC. The authors of the aforementioned studies also found that values of BIS, spectral edge frequency (SEF) 95% and median frequency (MED) were significantly higher during sevoflurane compared to isoflurane anaesthesia [14].

The ability of the BIS to predict motor responses during surgery in animals anaesthetized with ketamine/azaperone was also evaluated and compared to the electrically evoked nociceptive flexion reflex (NFR) [10]. With a prediction probability of $0.58 \pm 0.04$ for BIS and $0.76 \pm 0.03$ for NFR, the authors concluded that the NFR, but not the BIS, can be used for this purpose. On the contrary, another study found the BIS to have 96% sensitivity and 91% specificity for detecting a positive response following a mechanical stimulation (toe pinch) in isoflurane-anesthetised pigs [13].

Further EEG-based DoA indicators were found in the present review, being SEF (4/17 = 23.5%) [12, 14, 16, 18] and MED (3/40 = 42.9%) [12, 14, 16] among those used the most. In two studies [14, 16], SEF significantly decreased from baseline to 1 MAC, both with isoflurane, sevoflurane and desflurane, but not at higher concentrations; similar results were found for MED. Both DoA indicators were influenced by the presence of EEG suppression.

Four studies reported the use of raw EEG and spectral power [19–22]. Changes in EEG frequency and amplitude were reported while deepening the anaesthetic plane. Results of these studies also suggest that the EEG signal can be influenced not only by anaesthetic doses, but also by age, investigated brain regions and ventilation strategy (possibly linked to the arterial partial pressure of carbon dioxide levels).

The appearance of motor reactions following mechanical stimuli (nociceptive and non-nociceptive) was investigated as DoA indicator in 3/19 studies (15.8%), all using inhalant anaesthetics [24–26]. Different responses, depending on the stimulus applied, were observed. In particular, MAC values assessed by dew clamp were 19 and 21% higher compared to those obtained by tail clamp, when isoflurane or I-653 were used, respectively [24]. Moreover, no constant pattern of reaction appearance/disappearance following eyelashes brush, corneal brush, nasal septum pinch, interdigital fold pinch, periople pinch, tail and claw clamp was found at decreasing isoflurane concentrations [25]. In a recent study, the reaction to claw

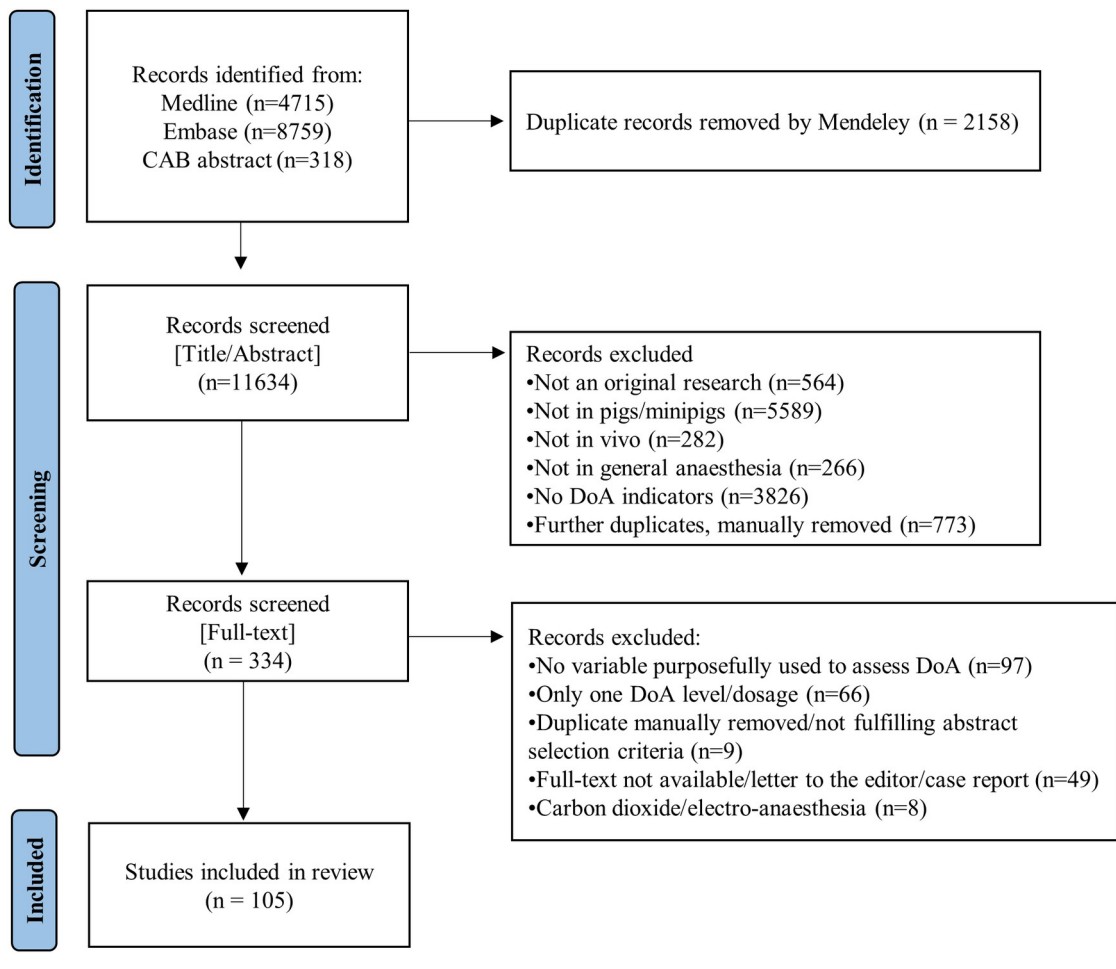

**Fig 1. Flow diagram of the study.**

clamp did not have a uniform pattern among animals when different drugs, at increasing doses, were administered in isoflurane anaesthetized pigs [26].

Electrical nociceptive stimuli were also applied in isoflurane anaesthetized pigs to evoke a nociceptive withdrawal reflex (NWR). Reflex amplitude decreased with increasing isoflurane concentrations, and reflexes evoked by single stimuli were suppressed at lower concentrations than those evoked by repeated stimuli, showing that temporal summation still occurs around

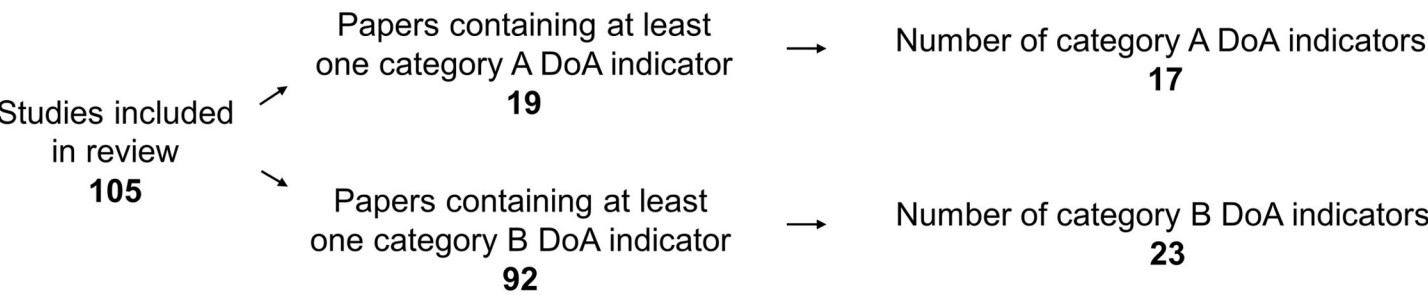

**Fig 2. Categorisation of the depth-of-anaesthesia (DoA) indicators.** The sum of the papers containing at least one category A or B DoA indicator exceeds the total number of studies included in the review, since some papers included both category A and category B DoA indicators.

**Table 1. Depth-of-anaesthesia indicators included in category A.**

| Depth of anaesthesia indicator | Number of papers | References |
|---|---|---|
| BIS[a] | 8 | [10–17] |
| SEF 95%[b] | 5 | [12, 14, 16, 18, 19] |
| Raw EEG[d] frequencies (power/visual) | 4 | [19–22] |
| MED[c] | 3 | [12, 14, 16] |
| NWR[f] | 3 | [10, 23] |
| Motor response/activity | 3 | [24–26] |
| VAS[e] | 2 | [12, 14] |
| IoC[g] | 1 | [27] |
| PSI[h] | 1 | [19] |
| Suppression ratio | 1 | [19] |
| Heart rate | 1 | [13] |
| Blood pressure | 1 | [13] |
| DAI[I] (based on middle-latency auditory evoked potentials) | 1 | [18] |
| COHmax[j] | 1 | [20] |
| aIF[k] | 1 | [20] |
| cIFmax[l] | 1 | [20] |
| MAC[m] | 1 | [28] |

[a] bispectral index;

[b] spectral edge frequency 95%;

[c] median frequency;

[d] electroencephalographic;

[e] visual analogue scale;

[f] nociceptive withdrawal reflex;

[g] index of consciousness;

[h] patient state index

[I] depth of anaesthesia index;

[j] coherence;

[k] auto-dependencies;

[l] cross-dependencies;

[m] minimum alveolar concentration.

MAC levels [23]. In a recent study, the NWR was shown to be influenced in a nonuniform manner by different drugs administered at increasing doses, and to give dissimilar results when compared to response to claw clamp [26].

**Category B.** All the studies focused on pigs and were published between 1980 and 2022.

The most reported DoA indicator was the appearance of a motor response following a stimulus (37/92 = 40.2%), that was nociceptive in 34/92 cases (37%). It was followed by DoA scores (21/92 = 22.8%), BIS (16/92 = 17.4%) and SEF 95% (9/81 = 9.8%).

Within the papers reporting DoA scores, 23 items were used to construct them, among which the most represented were: palpebral reflex (12/23 = 52.2%), reaction to painful stimuli (11/23 = 47.8%), body position/posture (11/23 = 47.8%), jaw tone (10/23 = 43.5%) and spontaneous movements (8/23 = 34.8%).

## Discussion

The present scoping review highlights the lack of scientifically valid methods to ensure an adequate DoA in pigs undergoing general anaesthesia. Only few investigations have been

**Table 2. Depth-of-anaesthesia indicators included in category B.**

| Depth of anaesthesia indicator | Number of papers | References |
|---|---|---|
| Motor response to a stimulus | 37 | [11, 29–64] |
| Score | 21 | [16, 53, 65–83] |
| BIS[a] | 16 | [52, 84–98] |
| SEF 95%[b] | 9 | [30, 64, 85, 86, 99–103] |
| EEG[c] suppression | 9 | [20, 25, 27, 85, 86, 104–107] |
| Palpebral reflex | 7 | [22, 41, 47, 48, 56, 59, 108] |
| Raw EEG frequencies (power/visual) | 5 | [85, 86, 99, 107, 109] |
| Corneal reflex | 4 | [27, 41, 47, 48] |
| Anal reflex | 2 | [47, 48] |
| Muscular tone | 2 | [42, 59] |
| MED[d] | 2 | [99, 110] |
| Pupil size | 2 | [22, 41] |
| Approximate entropy | 2 | [85, 86] |
| Permutation entropy | 2 | [85, 86] |
| CSI[e] | 1 | [111] |
| Narcotrend index | 1 | [55] |
| SSEP[f] | 1 | [112] |
| Average EEG amplitude | 1 | [106] |
| Nystagmus | 1 | [59] |
| Vocalization | 1 | [54] |
| TcMEP[g] | 1 | [113] |
| Conjunctival reflex | 1 | [41] |
| aEEG[h] | 1 | [114] |

[a] bispectral index;

[b] spectral edge frequency 95%;

[c] electroencephalographic;

[d] median frequency;

[e] cerebral state index;

[f] somatosensory evoked potentials;

[g] transcranial motor evoked potential;

[h] amplitude-integrated electroencephalography.

performed for this purpose. Moreover, heterogeneous and contrasting results about the usefulness of the DoA indicators used has been found, making difficult to draw final conclusions. Furthermore, inhalant anaesthetics have been mainly administered, and almost no information is present regarding DoA assessment during total intravenous anaesthesia.

Accidental awareness is one of the main consequences of an inadequate DoA. In humans, its incidence has been reduced from 1–2% in the 1980s to around 0.1% nowadays [7, 115, 116]. It has been previously highlighted that accidental awareness is likely to be prevalent in pigs undergoing experimental studies, mainly caused by inadequate monitoring and poorly trained staff [6]. This is in line with our results showing that, currently, no valid monitoring strategies are available, leading to an increased risk of accidental awareness. Such a lack, on one side impedes the development of DoA assessment guidelines, on the other side represents a great ethical issue of which the scientific community should be aware.

Consciousness vanishes when anaesthetics produce loss of cortical connectivity and block the brain's ability to integrate information. Following this principle, in the last two decades,

**Table 3. Category A depth-of-anaesthesia indicators, relative studies and summary of the main results.**

| Author Year | Age | Breed | Sex | Number of animals | Main anaesthetic | Depth-of-anaesthesia indicator | Summary of the main results |
|---|---|---|---|---|---|---|---|
| Baars 2013 [10] | ND[a] | ND[a] | Male | 30 | Ketamine | Nociceptive flexion reflex | The nociceptive flexion reflex could help predicting movements following a surgical stimulus. |
| | | | | | | BIS[b] | BIS[b] was not adequate to predict movements following a surgical stimulus. |
| Eger 1988 [24] | 1–4 months | (domestic) | ND[a] | 13 | 1–653 / isoflurane | Movements (tail clamp) | MAC[f] assessed by dew clamp was 19 and 21% greater compared to MAC[f] assessed by tail clamp, when isoflurane or I-653 was used, respectively. |
| | | | | | | Movement (claw clamp) | |
| Frasch 2007 [20] | 1–4 months | mixed German domestic breed | Female | 6 | Propofol | EEG[c] spectral power | Total EEG[c] power, as well as delta, theta and alpha frequency bands power, decreased in deep (burst suppression) versus moderate (30% of the propofol dose required to reach burst suppression) anaesthesia; no difference was found in spectral power over the temporoparietal region in the beta frequency band. |
| | | | | | | Coherence (COHmax) | Lower COHmax was found in deep versus moderate anaesthesia. |
| | | | | | | Auto-dependencies (aIF, complexity of information flow in ECoG[d] and EThG[e]) | The complexity of information flow over the cortical regions and in the reticulothalamocortical communication decreased during deep anaesthesia. |
| | | | | | | Cross-dependencies (cIFmax, maximum of complexity of information flow between cortico-cortical and reticulothalamocortical regions in ECoG[d] and EThG[e]) | Cross information flow max between reticulothalamocortical and frontoparietal, temporoparietal and parietooccipital regions increased in deep versus moderate anaesthesia. |
| Greene 2004 [11] | 1–4 months | Yorkshire-Landrace | Mix | 16 | Isoflurane | BIS[b] | BIS[b] appeared reliable for differentiating light versus deep anaesthesia, but of limited use at MAC[f]-multiples between 1 and 1.6. |
| Haga 2021 [26] | 1–4 months | Landrace-Duroc | Female | 6 | Isoflurane | Movement (claw clamp) | The two indicators did not give uniforms results when dexmedetomidine or fentanyl were administered at increasing doses. |
| | | | | | | Nociceptive withdrawal reflex | |
| Haga 2011 [28] | 1–4 months | Yorkshire-Landrace | Mix | 10 | Isoflurane | MAC[f] | No significant correlation was found between individual MAC[f] and burst suppression threshold values. |
| Haga 1999 [12] | ND[a] | Norwegian Landrace | Mix | 16 | Isoflurane | BIS[b] | BIS[b] did not accurately reflect anaesthetic depth. |
| | | | | | | VAS[s] | No linear trend was found between VAS[g] (depth of anaesthesia) and BIS[b]. |
| | | | | | | SEF 95%[h] | SEF 95%[h] was more sensitive than MED[i] in differentiating between consciousness and light levels of isoflurane anaesthesia. Burst suppression impeded SEF[h] and MED[i] calculation at deep anaesthetic levels. |
| | | | | | | MED[i] | |

*(Continued)*

**Table 3.** (Continued)

| Author Year | Age | Breed | Sex | Number of animals | Main anaesthetic | Depth-of-anaesthesia indicator | Summary of the main results |
|---|---|---|---|---|---|---|---|
| Haga 2002 [25] | ND[a] | Norwegian Landrace | Mix | 10 | Isoflurane | Movement (eyelashes brush) | No pig reacted to eyelash brush. |
| | | | | | | Movement (corneal brush) | At decreasing isoflurane concentrations (starting from presence of EEG[c] burst suppression), none of the stimuli consistently increased the magnitude of response, that moreover was inconsistent among pigs. Motor response to each stimulus (except eyelash brushing) occurred in at least one pig during EEG[c] suppression. |
| | | | | | | Movement (nasal septum pinch) | |
| | | | | | | Movement (anus pinch) | |
| | | | | | | Movement (interdigital skin fold pinch) | |
| | | | | | | Movement (periople pinch) | |
| | | | | | | Movement (tail clamp) | |
| | | | | | | Movement (claw clamp) | |
| Jaber 2015 [13] | ≤1 month | Yorkshire-cross | Female | 33 | Isoflurane | Heart rate | At a stable isoflurane concentration, piglets having a positive response to toe pinch showed a significant increase in heart rate but not in blood pressure. However, the receiver operating characteristic (ROC) analysis revealed that heart rate was not a sensitive predictor of toe pinch motion response. Both before and after toe pinch, BIS[b]-mean was a strong predictor of a positive motion response. |
| | | | | | | Blood pressure | |
| | | | | | | BIS[b] | |
| Llonch 2011 [27] | ND[a] | Landrace-Large white-Pietrain | ND[a] | 6 | Propofol | IoC[j] | IoC followed the deepening of the anaesthesia level. |
| Martín-Cancho 2006 [16] | ND[a] | Large white-Landrace | Mix | 16 | Desflurane | BIS[b] | BIS[b], but not SEF[h], MED[i] and the simple descriptive scale (SDS) used, was deemed useful to predict changes in anaesthetic depth. |
| | | | | | | SEF 95%[h] | |
| | | | | | | MED[i] | |
| Martín-Cancho 2004 [15] | 1–4 months | Large White-Landrace | Mix | 12 | Sevoflurane / Propofol | BIS[b] | BIS[b] was not useful for predicting the speed of recovery from anaesthesia and the clinically important changes in arterial blood pressure and heart rate. |
| Martín-Cancho 2003 [14] | ND[a] | Large white-Landrace | Mix | 32 | Isoflurane, Sevoflurane | BIS[b] | BIS[b] was useful for predicting changes in anaesthetic depth at clinical doses of both isoflurane and sevoflurane. Values of BIS[b], SEF[h] and MED[i] were significantly higher while sevoflurane was administered compared to isoflurane, at equivalent MACs[f]. A good correlation between VAS[g] and BIS[b] and between VAS[g] and SEF[h] was present. |
| | | | | | | VAS[g] | |
| | | | | | | SEF 95%[h] | |
| | | | | | | MED[i] | |
| Martoft 2001 [18] | ND[a] | Danish Landrace-Duroc | Mix | 6 | Thiopentone | Depth of anaesthesia index (based on middle-latency auditory evoked potentials (MLAEP)) | During anaesthesia induction, MLAEP latencies increased while MLAEP amplitudes and SEF 95%[h] decreased. |
| | | | | | | SEF 95%[h] | |
| Mirra 2022 [19] | 1–4 months | Phenotypic Edelschwein | Mix | 5 | Isoflurane | PSI[k] | Modification in EEG[c] frequency bands power and PSI[k] seemed to reflect different anaesthetic phases. A less clear behavior was found for SEF 95%[h] and suppression ratio. |
| | | | | | | SEF 95%[h] | |
| | | | | | | Suppression ratio | |
| | | | | | | raw EEG[c] | |

(*Continued*)

**Table 3.** (Continued)

| Author Year | Age | Breed | Sex | Number of animals | Main anaesthetic | Depth-of-anaesthesia indicator | Summary of the main results |
|---|---|---|---|---|---|---|---|
| Rose 1972 [21] | 1–4 months | Landrace Large White | Female | 51 | Halothane | raw EEG[c] | Differences in amplitude and frequency were noticed in the raw EEG[c] signal between neonatal and older pigs, and between ventilated and not ventilated pigs (possibly due to different arterial partial pressure of carbon dioxide levels). Fast EEG[c] rhythms were always dominant (12–14 and 24-26Hz), often associated with a background slower activity (8–12 Hz). |
| Schmidt 2000 [17] | ND[a] | German Landrace | ND[a] | 8 | Halothane, Xenon, Pentobarbital | BIS[b] | BIS[b] levels during halothane and xenon anaesthesia were comparable to those measured under total intravenous anaesthesia with pentobarbital. |
| Schneider 1980 [22] | 1–4 months / 4–6 months | Veredeltes Landschwein | ND[a] | 12 | Methitural | raw EEG[c] | Excitation during induction was accompanied by an increase in beta waves, while the deepening of the anaesthetic plane by an increase in theta and alpha waves; theta waves persisted during the whole anaesthetic time. During the recovery there was an increase in delta, alpha and beta waves. |
| Spadavecchia 2012 [23] | 1–4 months | Yorkshire–Landrace | Mix | 10 | Isoflurane | Nociceptive withdrawal reflex | Nociceptive withdrawal reflex amplitude correlated with isoflurane concentration. The stimulation paradigm influenced the responses. |

[a] not declared;

[b] bispectral index;

[c] electroencephalography;

[d] electrocorticogram;

[e] electrothalamogram;

[f] minimum alveolar concentration;

[g] visual Analogue Scale;

[h] spectral edge frequency 95%;

[i] median frequency;

[j] index of consciousness;

[k] patient state index.

many investigations have been conducted to assess brain activity as DoA indicator. The modification of EEG amplitude and frequency over time reflects drugs action on brain circuits, until total suppression of the EEG signal is reached. Being the EEG real-time visual interpretation difficult, algorithms have been developed and included in EEG-based monitors, in order to provide the anaesthetists with easy-to-follow DoA indexes. In the last decades, many have been integrated in medical devices. Among them, only Index of Consciousness (IoC), Patient State Index (PSI) and BIS have been evaluated as DoA indicators in pigs. For IoC [27] and PSI [19], only single reports could be found; both indexes seemed to mirror the clinical judgement in animals either receiving a single propofol bolus or undergoing an isoflurane-based balanced anaesthesia, respectively. Further studies are needed to confirm these preliminary results. On the other side, the BIS has been assessed in eight studies (seven of which using inhalants) and contrasting results were reported. Greene and colleagues [11] and Haga and colleagues [12] found that, at increasing doses of isoflurane, the BIS remained stable, not providing any help

in differentiating anaesthetic levels. These results are in contrast with those of Martín-Cancho and colleagues that showed how BIS could reflect increasing concentrations of isoflurane, sevoflurane and desflurane [14, 16]. Schmidt and colleagues [17] reported similar BIS values while halothane, xenon or pentobarbital were administered, but various levels of DoA were not compared. Jaber and colleagues [13] reported that BIS can possibly be used to predict movements in response to noxious stimulation, in contrast with the results found by Baars and colleagues [10]. If the BIS has been developed to linearly follow an increasing DoA or rather to provide a range of values corresponding to an adequate DoA is not known. Indeed, the algorithm through which the EEG signal is processed is proprietary, and conclusions are difficult to be drawn.

It is also important to highlight that BIS, IoC and PSI have been developed based on data collected from humans, and a mere translation of their application in pigs could lead to misleading results. Species-specific anatomical peculiarities should be considered [19]. Furthermore, age, drugs used, clinical status, among others, have also been demonstrated to influence the EEG signal, in both humans and pigs [21, 22, 117, 118] but are often not taken in consideration by the EEG-based DoA monitors developed so far. Further studies should be conducted to assess the validity of the BIS and other indexes in assessing DoA in pigs, using various anaesthetic regimens.

Other features extrapolated from the EEG have been applied to assess DoA in this species. Among them, SEF 95% and MED have been mostly reported. They represent the frequency below which 95% and 50% of the EEG total power is located, respectively. Both parameters were found not capable to correctly predict DoA in pigs. Moreover, they were largely influenced by the type of anaesthetic used, as well as by the presence of EEG suppression, similarly to what reported in humans [119–121]. Also in this case, no studies using total intravenous anaesthesia have been performed. Further investigations should be carried out before drawing final conclusions.

The use of EEG suppression to reach an adequate DoA has also been reported in pigs, but never thoroughly investigated. The suppression ratio represents the percentage of EEG activity that does not exceed a user-specified amplitude threshold (usually $\pm 5\,\mu V$) over a certain time span (usually 60–63 seconds) [122, 123]. No specific end-points have been established in pigs, but thresholds values of 20% or above has been targeted to reach deep anaesthesia. Recent studies in humans [123, 124] found the intraoperative presence of EEG suppression to be an independent risk factor for postoperative delirium. Despite no strong causative effect has been proved, and no information is available on the influence that EEG suppression might have on recovery outcomes in veterinary species, care should be taken in titrating drug administration only based on EEG suppression levels. Moreover, as previously mentioned, caution should be used in quantifying EEG suppression via DoA monitors whose algorithms have been created for humans.

Despite the mechanism of action of general anaesthetics has not been fully understood, their action is observed both at cortical and subcortical levels [125–127]. While the consciousness is mainly based on cortical neuronal networks, nociception and related motor responses are rather processed at subcortical levels [128–132].

Assessing whether a nociceptive stimulus is able to evoke a reaction or not was found to be the method mostly used to evaluate DoA in pigs. On responsiveness to stimulation is based the concept of MAC, namely the minimum alveolar concentration of anaesthetic at 1 atmosphere, which produces immobility in 50% of subjects exposed to a noxious stimulus [133]. It has often been used as a guide for inhalant drugs administration in many animal species. However, if consciousness is a state in which an individual is able to process information from his surroundings [134], MAC seems to be mainly related to spinal cord excitability rather than to

consciousness [128–132]. This is in line with studies in humans, finding patient's responsiveness and consciousness to be not always related to each other [135, 136]. Thus, the use of MAC as sole method to assess DoA should be avoided or at least its limitations clearly recognised.

The electrical NWR has also been investigated as DoA indicator. Even if this methodology has been principally applied to assess antinociception, studies support its possible use as adjunct in the DoA assessment [10, 23]. Further investigations should be carried out to establish the most appropriate recording paradigms in this species, and to evaluate the effect that different drugs have on it.

Depth of anaesthesia scores have been widely used in the literature in pigs. However, none has been validated, and the scoring has been often based on the investigator subjective judgment. Despite being easy to apply, care should be taken when assessing DoA using scores which have not undergone a validation process.

Great effort has been put in the last years for the development, diffusion and promotion of the 3Rs principle [137–139]. One of its three cardinal points is the "refinement", namely the minimisation of pain, suffering, distress or lasting harm that may be experienced by research animals. In spite of that, only two studies have been performed to investigate DoA indicators in pigs in the last seven years.

All the pigs included in the selected papers were younger than 6 months of age. Even if juvenile animals are commonly employed in experimental settings due to the ease of handling, no information is currently available on any DoA indicator in adult and geriatric pigs. Care should be taken while translating the information currently available on DoA assessment to animals of different ages.

No information has been found in minipigs, despite their increasing use in different branches of translational medicine [140–143]. Further specific studies should be performed characterizing the effects induced by anaesthetic drugs in these animals, also considering that most of them are used as adult.

Considering the nature of our research question, a scoping review was deemed the most appropriate design. Due to the heterogeneity in the included studies, no risk of bias assessment was performed. Our results can be the basis for future focused systematic reviews, aiming at answering narrower and more specific questions and analysing the quality of individual studies.

Our scoping review has some limitations. First, due to limited personnel availability, not all the screening steps were performed by two persons. Despite the guidelines do not establish a minimum number of investigators [8], the involvement of more than one independent reviewer during the whole process would have guaranteed a higher level of reproducibility. Second, despite following a rigorous methodology, it is possible that relevant studies were missed during abstract selection if neither the title nor the abstract mentioned any DoA indicator.

## Conclusion

No appropriate DoA indicator is currently available for pigs and minipigs, and welfare during anaesthesia cannot be fully ensured. Validation studies should be performed on already available and new DoA indicators, in order to provide valid and reliable tools to the scientific community, to be applied in experimental settings, especially during invasive procedures.

## Supporting information

**S1 File. Scoping review protocol.**
(DOCX)

**S2 File. Search strings developed for the literature research.**
(DOC)

**S3 File. Abstract screening process.**
(XLSX)

**S4 File. Full-text data extraction tables.**
(XLSX)

**S5 File. PRISMA-ScR checklist.**
(DOCX)

# Acknowledgments

The authors thank Heidrun Janka, the anaesthetists of the Anaesthesiology and Pain Therapy Section of the Vetsuisse Faculty-University of Bern, and the ACVA-List members who replied to our email, for the support given in the search strategy development. The authors thank Shannon Axiak Flammer for her contribution in improving the manuscript.

# Author Contributions

**Conceptualization:** Alessandro Mirra, Luís Pedro Carmo, Olivier Levionnois, Claudia Spadavecchia.

**Data curation:** Alessandro Mirra, Ekaterina Gamez Maidanskaia.

**Formal analysis:** Alessandro Mirra, Ekaterina Gamez Maidanskaia, Olivier Levionnois, Claudia Spadavecchia.

**Investigation:** Alessandro Mirra, Ekaterina Gamez Maidanskaia, Luís Pedro Carmo, Olivier Levionnois, Claudia Spadavecchia.

**Methodology:** Alessandro Mirra, Ekaterina Gamez Maidanskaia, Luís Pedro Carmo, Olivier Levionnois, Claudia Spadavecchia.

**Supervision:** Luís Pedro Carmo, Olivier Levionnois, Claudia Spadavecchia.

**Validation:** Luís Pedro Carmo, Olivier Levionnois, Claudia Spadavecchia.

**Visualization:** Luís Pedro Carmo.

**Writing – original draft:** Alessandro Mirra.

**Writing – review & editing:** Ekaterina Gamez Maidanskaia, Luís Pedro Carmo, Olivier Levionnois, Claudia Spadavecchia.

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
