## [Decision Letter · Decision Letter 0]

26 Jan 2023

PONE-D-22-28207How is depth of anaesthesia assessed in experimental pigs? A scoping reviewPLOS ONE

Dear Dr. Mirra,

Thank you for submitting your manuscript to PLOS ONE. After careful consideration, we feel that it has merit but does not fully meet PLOS ONE’s publication criteria as it currently stands. Therefore, we invite you to submit a revised version of the manuscript that addresses the points raised during the review process.

ACADEMIC EDITOR:please assess all the reviewers comments 

We look forward to receiving your revised manuscript.

Kind regards,

Silvia Fiorelli

Academic Editor

PLOS ONE

Journal Requirements:

Reviewers' comments:

Reviewer's Responses to Questions

**Comments to the Author**

1. Is the manuscript technically sound, and do the data support the conclusions?

Reviewer #1: Yes

Reviewer #2: Yes

2. Has the statistical analysis been performed appropriately and rigorously? 

Reviewer #1: Yes

Reviewer #2: Yes

3. Have the authors made all data underlying the findings in their manuscript fully available?

Reviewer #1: Yes

Reviewer #2: Yes

4. Is the manuscript presented in an intelligible fashion and written in standard English?

Reviewer #1: Yes

Reviewer #2: Yes

5. Review Comments to the Author

Reviewer #1: well-conducted review of literature and analyze of different monitors and parameters. Constructive clear discussion to replace the contexte of avoid of awareness and memorization. the work takes place in the background of translational research using pigs

Reviewer #2: ABSTRACT

Line 45: It should be made clearer that the contrast results are not in relation to how useful the depth of anaesthesia indicators are but to what are the best depth of anaesthesia indicators.

Lines 50-51: Maybe the authors should underline the lack of many high quality scientific studies and not only the lack of scientifically valid and reliable indicators of depth anaesthesia in pigs.

INTRODUCTION

Line 81: Why do they claim that the reliability of the results obtained from translational studies is questioned by the lack of valid depth of anaesthesia indicators? It should be explained in more detail.

Lines 87-88: Maybe the answer to this question is already elucidated in the lines 80-81, so the question should not address the usefulness of methodologies to assess DoA in pigs but should refer the best validated ones.

Lines 89-91: It is not necessary to describe the differences between a scoping review and a systematic review in the introduction. The authors could explain why they chose a scoping review analysis in the discussion.

MATERIAL AND METHODS

Lines 170-172: During the data extraction it is very important to know the different quality of the studies evaluated in order to give more emphasis to the high quality studies as well as to show the current gaps in scientific literature.

RESULTS

Lines 287-288: This result has not been sufficiently clarified, references 11 and 12 are quite different in their methodology and results, hence it should be analysed more in depth.

Lines 288: The studies cited, references 11-12-14-16, are not so similar, therefore this statement appears to be incomplete. BIS was compared to different variables with different results and it should be explained accurately.

DISCUSSION

Lines 364: Perhaps the results about using IOC and PSI indexes should be expressed in detail and discussed.

Lines 365-371: The BIS method is evaluated in 8 studies but the authors only discuss 5 out 8; they should also discuss the remaining three studies. Besides, the authors could give some possible reasons for the contrasting results when using the BIS method.

6. PLOS authors have the option to publish the peer review history of their article (what does this mean?). If published, this will include your full peer review and any attached files.

Reviewer #1: No

Reviewer #2: **Yes: **Massimiliano Pelli

---

## [Author Response · Author response to Decision Letter 0]

9 Feb 2023

RESPONSE TO REVIEWERS

Dear reviewers,

First, thank you very much for the time dedicated to our manuscript, as well as for the suggestions made to improve its quality.

Please find below the answers to your comments.

Reviewer #1: well-conducted review of literature and analyze of different monitors and parameters. Constructive clear discussion to replace the contexte of avoid of awareness and memorization. the work takes place in the background of translational research using pig

Thank you, we appreciate it.

Reviewer #2: 

Thank you for the rigorous assessment of our manuscript and the suggestions provided.

ABSTRACT

Line 45: It should be made clearer that the contrast results are not in relation to how useful the depth of anaesthesia indicators are but to what are the best depth of anaesthesia indicators.

Thank you for your comment. Contrasting results have been found in relation to how useful the depth of anaesthesia (DoA) indicators were. We did not find papers specifically analysing which indicator was the best to assess DoA in pigs. The sentence has been modified to clarify better our results.

Lines 50-51: Maybe the authors should underline the lack of many high quality scientific studies and not only the lack of scientifically valid and reliable indicators of depth anaesthesia in pigs.

Thank you for pointing it out. Due to the nature of the studies found and the type of review chosen, we did not analyse the quality of the papers, but rather screened their results. Our scoping review could be the base for future focused systematic reviews, that could look at the quality of single studies. We added a sentence in the discussion section

INTRODUCTION

Line 81: Why do they claim that the reliability of the results obtained from translational studies is questioned by the lack of valid depth of anaesthesia indicators? It should be explained in more detail.

Thank you for spotting it. A sentence has been added to better explain the concept. The lack of knowledge on how to appropriately assess DoA brings to the difficulty of ensuring a correct DoA. This refers not only to the possible occurrence of intraoperative awareness per se, but also to all the correlated consequences (e.g., cardiovascular, respiratory, hormonal). This brings to the impossibility to differentiate, for example, physiological modifications occurring because of a certain surgical stimulation/treatment from those occurring due to an inappropriate DoA/intraoperative awareness.

Lines 87-88: Maybe the answer to this question is already elucidated in the lines 80-81, so the question should not address the usefulness of methodologies to assess DoA in pigs but should refer the best validated ones.

This comment has been partially answered above. Unfortunately, in veterinary medicine and especially in pigs, no validated DoA indicators are present (as also shown by the present review). Thus, we did not want to look only for validated DoA indicators, but rather for all the methodologies used in pigs so far.

For example, we found that reaction to claw clamp has been widely used as sole DoA indicator, despite being inappropriate for this scope when used alone. Our review highlights how DoA assessment has not been always done in a correct way in pigs, and how really few steps toward a better DoA assessment have been performed in the last decades.

Lines 89-91: It is not necessary to describe the differences between a scoping review and a systematic review in the introduction. The authors could explain why they chose a scoping review analysis in the discussion.

Thank you for your suggestion. The sentence has been removed from the Introduction and a small paragraph has been added in the discussion section.

MATERIAL AND METHODS

Lines 170-172: During the data extraction it is very important to know the different quality of the studies evaluated in order to give more emphasis to the high quality studies as well as to show the current gaps in scientific literature.

Even if it would be really interesting to analyse the quality of single studies, this was out of the scope of the present review. Unfortunately, the nature of the published papers did not allow us to perform a systematic review (for which a risk of bias assessment is a requirement) and/or to have a homogenous assessment of the quality of single studies.

RESULTS

Lines 287-288: This result has not been sufficiently clarified, references 11 and 12 are quite different in their methodology and results, hence it should be analysed more in depth.

Thank you for your suggestion. We modified the text focusing on the outcome that was of most interest for us, meaning the statistical difference among BIS values at clinically relevant DoA levels (identified in all the studies as MAC and MAC-multiples). 

Lines 288: The studies cited, references 11-12-14-16, are not so similar, therefore this statement appears to be incomplete. BIS was compared to different variables with different results and it should be explained accurately.

Linked to the previous comment: we slightly modified the text to let the reader understand that we took in consideration only the outcome that was in common among studies and of interest for us (statistical difference among DoA levels, identified with MAC and MAC-multiples).

DISCUSSION

Lines 364: Perhaps the results about using IOC and PSI indexes should be expressed in detail and discussed.

Thank you for spotting it. A sentence has been added to provide further information.

Lines 365-371: The BIS method is evaluated in 8 studies but the authors only discuss 5 out 8; they should also discuss the remaining three studies. Besides, the authors could give some possible reasons for the contrasting results when using the BIS method.

Thank you for your suggestion. We agree with you; the other three studies have been included in the discussion. Possible reasons for the heterogenicity of the results and the unsuitability of BIS in pigs are discussed in the following lines.

---

## [Decision Letter · Decision Letter 1]

9 Mar 2023

How is depth of anaesthesia assessed in experimental pigs? A scoping review

PONE-D-22-28207R1

Dear Dr. Mirra,

We’re pleased to inform you that your manuscript has been judged scientifically suitable for publication and will be formally accepted for publication once it meets all outstanding technical requirements.

Kind regards,

Silvia Fiorelli

Academic Editor

PLOS ONE

Additional Editor Comments :

Congratulations to the authors and thanks to the reviewers for the suggestions provided which really helped improve the quality of the manuscript.

Reviewers' comments:

Reviewer's Responses to Questions

**Comments to the Author**

1. If the authors have adequately addressed your comments raised in a previous round of review and you feel that this manuscript is now acceptable for publication, you may indicate that here to bypass the “Comments to the Author” section, enter your conflict of interest statement in the “Confidential to Editor” section, and submit your "Accept" recommendation.

Reviewer #1: All comments have been addressed

Reviewer #2: All comments have been addressed

2. Is the manuscript technically sound, and do the data support the conclusions?

Reviewer #1: Yes

Reviewer #2: Yes

3. Has the statistical analysis been performed appropriately and rigorously? 

Reviewer #1: Yes

Reviewer #2: N/A

4. Have the authors made all data underlying the findings in their manuscript fully available?

Reviewer #1: Yes

Reviewer #2: Yes

5. Is the manuscript presented in an intelligible fashion and written in standard English?

Reviewer #1: Yes

Reviewer #2: Yes

6. Review Comments to the Author

Reviewer #1: The precisions that have been added improve the manuscript. In the abstract, the conflicting results in DoA indicators usefulness appear more clearly.

Some studies cited use stimulus-reponse as a DoA indicator evaluation. A short explanation on difference, crucial in humans'care, between awareness and recall and movement to surgical stimulation could be added in the discussion.

Minipigs without any study found appear the same way as pigs in the conclusion, I would find better to separate theses different situations with absence of study or absence of proof in the studies

Reviewer #2: All my comments were correctly evalueted and analyzed.

You did appropriated modifications of the text.

7. PLOS authors have the option to publish the peer review history of their article (what does this mean?). If published, this will include your full peer review and any attached files.

Reviewer #1: No

Reviewer #2: **Yes: **massimiliano pelli

---

## [Editor Report · Acceptance letter]

14 Mar 2023

PONE-D-22-28207R1 

How is depth of anaesthesia assessed in experimental pigs? A scoping review 

Dear Dr. Mirra:

I'm pleased to inform you that your manuscript has been deemed suitable for publication in PLOS ONE. Congratulations! Your manuscript is now with our production department. 

Kind regards, 

on behalf of

Dr. Silvia Fiorelli 

Academic Editor

PLOS ONE